# Tunable inertia of chiral magnetic domain walls

Jacob Torrejon[1,2], Eduardo Martinez[3] & Masamitsu Hayashi[1,4]

The time it takes to accelerate an object from zero to a given velocity depends on the applied force and the environment. If the force ceases, it takes exactly the same time to completely decelerate. A magnetic domain wall is a topological object that has been observed to follow this behaviour. Here we show that acceleration and deceleration times of chiral Neel walls driven by current are different in a system with low damping and moderate Dzyaloshinskii–Moriya exchange constant. The time needed to accelerate a domain wall with current via the spin Hall torque is much faster than the time it needs to decelerate once the current is turned off. The deceleration time is defined by the Dzyaloshinskii–Moriya exchange constant whereas the acceleration time depends on the spin Hall torque, enabling tunable inertia of chiral domain walls. Such unique feature of chiral domain walls can be utilized to move and position domain walls with lower current, key to the development of storage class memory devices.

[1] National Institute for Materials Science, Tsukuba 305-0047, Japan. [2] Unité Mixte de Physique CNRS/Thales, 1 Avenue Augustin Fresnel, 91767 Palaiseau, France. [3] Departamento de Fisica Aplicada, University of Salamanaca, Plaza de los Caidos s/n, E-37008 Salamanca, Spain. [4] Department of Physics, The University of Tokyo, Bunkyo, Tokyo 113-0033, Japan. Correspondence and requests for materials should be addressed to M.H. (email: hayashi@phys.s.u-tokyo.ac.jp).

It is now well established that a magnetic domain wall (DW) can be considered a topological object with effective mass[1–5] and momentum. For such an object, it requires certain time to accelerate right after a stimuli is turned on and to decelerate once the stimuli is removed. According to a model used to describe DWs, the acceleration and deceleration times of a DW are defined by the same material parameters that include the Gilbert damping constant, saturation magnetization and the dimension of the magnetic wire. The acceleration and deceleration times of a DW have been found to be the same when the DW is driven by current[6] via the spin transfer torque (STT) or by magnetic field[7,8]. Under such circumstances the distance a DW travels scales with the pulse length. Experimentally, identical acceleration and deceleration times manifest itself as a pulse length-independent quasi-static velocity[6,9], a measure of speed obtained in experiments by dividing the total distance the DW travelled during and after the pulse application with the pulse length.

Recent reports have shown that chiral Neel DWs[10] emerge owing to the Dzyaloshinskii–Moriya (DM) interaction at interfaces of magnetic layer and a heavy metal layer with strong spin–orbit coupling[11–22]. Such chiral Neel walls can be driven[23,24] by current via the spin Hall torque that arises when spin current is generated by the spin Hall effect in the heavy metal layer and diffuses into the magnetic layer[25–27].

Here we find that the quasi-static velocity of current (that is, spin Hall torque) driven chiral DWs increases as the current pulse length is reduced, indicating that the distance a DW travels does not scale linearly with the pulse length. The change in the quasi-static velocity with pulse length depends on the current passed along the film plane as well as the film stack. Using collective coordinate and full micromagnetic models, we show that the deceleration time is significantly longer than the acceleration time, giving rise to a driving force-dependent tunable inertia.

## Results

**Pulse length-dependent quasi-static domain wall velocity.** The film stack studied is Si-sub/W($d$)/Co$_{20}$Fe$_{60}$B$_{20}$(1)/MgO(2)/Ta(1) (units in nanometers). Two film sets (A and B) with nominally the same film structure are made and evaluated. The magnetic and transport properties of the two sets are slightly different (see the Methods section and Supplementary Table 1). We study wires with width ($w$) of $\sim$5 and $\sim$50 μm patterned from the films. An optical microscopy image of a representative $\sim$50 μm wide wire is shown in Fig. 1i inset together with the definition of the coordinate axis. Positive current corresponds to current flow along $+x$. Magneto-optical Kerr microscopy is used to measure the quasi-static velocity ($v_{END}$) of the DW. Positive velocity indicates that the DW moves to $+x$. (See Supplementary Note 1; Supplementary Figs 1 and 2 for the pulse transmission characterisitcs of a typical device).

Figure 1a–f shows the wall velocity as a function of pulse amplitude for films with different $d$. The pulse length ($t_P$) is fixed to 10 ns. The DWs move along the current flow regardless of the wall type ($\downarrow\uparrow$ and $\uparrow\downarrow$ walls). For current pulses with amplitude larger than the depinning threshold, the velocity increases with increasing pulse amplitude and eventually saturates. Such trend is consistent with the DW velocity driven by the spin Hall torque[23],

$$v = v_D \Big/ \sqrt{1 + \left(\frac{J_D}{J - J_C}\right)^2} \qquad (1)$$

where $v_D = \gamma \Delta H_{DM}$ is the saturation velocity and $J_D = \alpha J H_{DM}/H_{SH}$ is the current density at which the velocity saturates. $H_{DM} = \frac{D}{\Delta M_s}$ is the DM exchange field and $H_{SH} = -\frac{\hbar \theta_{SH}}{2eM_s t_{FM}}J$ is the damping-like effective field due to the spin Hall torque. Here $\gamma$ is the gyromagnetic ratio, $e$ is the electric charge, $\hbar$ is the reduced Planck constant. $\alpha$ is the Gilbert damping constant, $M_s$ is the saturation magnetization, $\Delta$ is the DW width and $t_{FM}$ is the thickness of the magnetic layer. $\theta_{SH}$ is the spin Hall angle of the heavy metal (W) layer and $D$ is the DM exchange constant. We have added an empirical threshold current density $J_C$ to equation (1) to account for the pinning. Note that equation (1) does not take into account transient effects which can influence the estimation of the wall velocity[28]. However, same results are also obtained by numerical solving the one-dimensional (1D) collective coordinate model[1], which naturally accounts for pinning and transient effects (Supplementary Fig. 3).

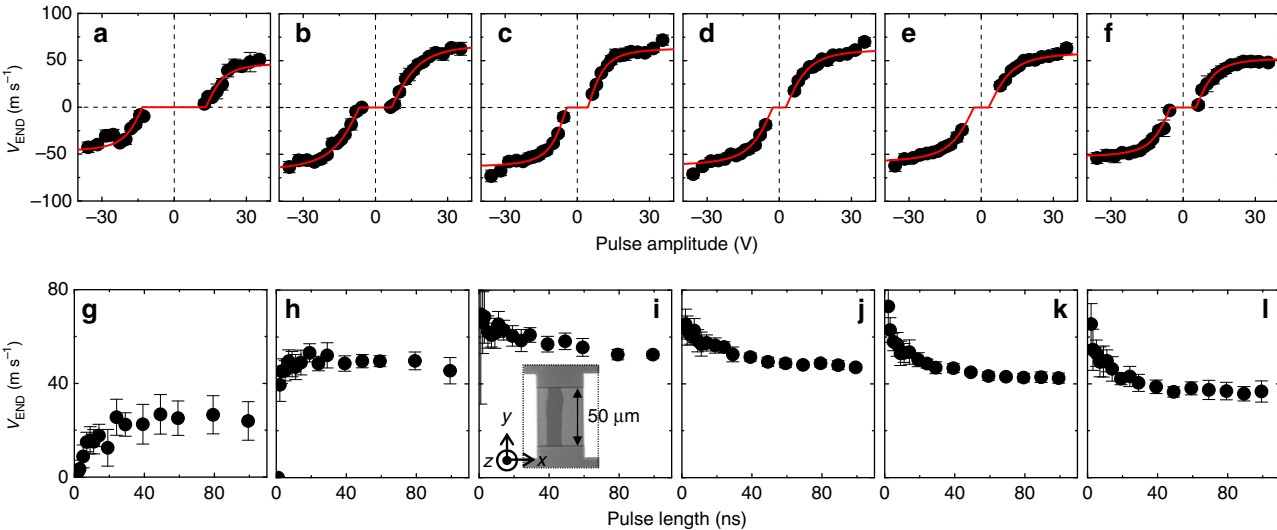

**Figure 1 | Pulse amplitude and pulse length-dependent domain wall (DW) velocity.** (a–f) Quasi-static DW velocity $v_{END}$ plotted against pulse amplitude for a fixed pulse length ($t_P = 10$ ns). The red solid line represents fitting with the 1D model (equation (1)). (g–l) Pulse length dependence of $v_{END}$ for fixed pulse amplitude ($\pm 16$ V). Symbols represent the average $|v_{END}|$ for both positive (16 V) and negative ($-16$ V) pulse amplitudes. The W layer thickness $d$ varies for **a–f** and **g–l** as 2.3, 2.6, 3.0, 3.3, 3.6, 4.0 nm. Inset of **i**: representative optical (Kerr) microscopy image of the device and the definition of the coordinate axis. All results are from film set A, wire width is $\sim$50 μm. The error bars represent standard deviation of the velocity estimated in three sections of the wire (see Methods for the definition of the sections).

The red solid lines in Fig. 1a–f show fitting of the experimental data using equation (1). Except for the thinnest W layer device, we find that the saturation velocity decreases when the W layer thickness ($d$) is increased. The corresponding $t_P$ dependence of $v_{END}$ for each device is plotted in Fig. 1g–l. For the thick W underlayer films, $v_{END}$ increases with decreasing pulse length. This is particularly evident when $t_P$ is shorter than $\sim 10$–20 ns. These results show that the distance a DW travels does not linearly scale with the pulse length, which is in striking difference with the STT driven DWs[6,8,9] or current driven narrow DWs in large magnetic damping system[29,30]. In contrast, $v_{END}$ drops for shorter pulses when the thickness of W is reduced below $\sim 3$ nm. See Supplementary Figs 4 and 5 for supporting experimental results.

**The one-dimensional model of domain walls.** To clarify the origin of the pulse length-dependent velocity, the dynamics of chiral DWs under current pulses are studied using the 1D collective coordinate model[1] with the spin Hall torque and the DM interaction included. The wall dynamics is described using three time-dependent variables: the wall position $q(t)$, the wall magnetization angle $\psi(t)$ and the tilting angle of the wall normal $\chi(t)$ (refs 28,31,32): see inset of Fig. 2a for the definition of the angles. Typical parameters of W/CoFeB/MgO (see Fig. 2 caption) are used and here we consider only the damping-like component of the spin Hall torque[26,27]. Using micromagnetic simulations we

find that the presence of any field-like torque has little impact on the relaxation times which are discussed later.

The numerically calculated temporal evolution of the wall velocity $v(t)$, the magnetization angle $\psi(t)$ and the tilting angle $\chi(t)$ under current pulses with fixed amplitude ($J = 0.5 \times 10^8$ A cm$^{-2}$) and length ($t_P \sim 100$ ns) are shown in Fig. 2a–c for an ideal wire with no pinning. Note that $v(t)$ is the instantaneous velocity at time $t$ and is different from $v_{END}$. Two extreme damping values, $\alpha = 0.01$ (black solid line) and $\alpha = 0.3$ (red dashed line), are used to illustrate the transient effects.

There are two distinct features that are characteristics of spin Hall torque driven chiral DWs. First, the acceleration time (or the rise time) and the deceleration time (or the fall time) of the wall velocity are significantly different for the low damping system (Fig. 2a, black solid line): the acceleration time is much faster than the deceleration time. Such effect is significantly suppressed when the Gilbert damping constant is larger[29,30] (Fig. 2a, red dashed lines). Note that the acceleration/deceleration times of the velocity are correlated with those of the wall magnetization angle $\psi(t)$, see Fig. 2a,b.

To provide a qualitative understanding, we analytically solve the differential equations of the 1D model using a linear approximation for a rigid wall ($\chi(t) = 0$). The analytical expression of the acceleration time ($\tau_A$) and deceleration time ($\tau_D$) reads (Supplementary Note 2):

$$\tau_A = \frac{1 + \alpha^2}{\gamma \left| \alpha H_K + \frac{\pi}{2} H_{SH} \right|} \qquad (2)$$

$$\tau_D = \frac{1 + \alpha^2}{\gamma \alpha \left| -H_K + \frac{\pi}{2} H_{DM} \right|} \qquad (3)$$

where $H_K = \frac{4 t_{FM} M_s \log(2)}{\Delta}$ is the magneto-static anisotropy field associated with the wall[28,33]. Equations (2) and (3) explicitly show the difference of the two quantities. The acceleration time depends on the spin Hall torque $H_{SH}$ (and therefore the current density) whereas the deceleration time is dependent on the DM exchange field $H_{DM}$. In the absence of the spin Hall torque and the DM exchange field, $\tau_A = \tau_D = \frac{1 + \alpha^2}{|\alpha \gamma H_K|}$, which has been derived for the STT driven DWs[6]. Note that $\tau_{A(D)}$ evolves during the transient process (that is, right after the current is turned on and off) and the relaxation times here represent the corresponding values when the angle magnetization is close to Bloch ($\tau_A$) or Neel ($\tau_D$) configurations. See Supplementary Note 2 and Supplementary Figs 6 and 7 for discussion on the linearized 1D model.

The second characteristic feature of Fig. 2a–c is the non-negligible drop in the wall velocity after the current pulse is turned on. Such drop in the wall velocity only occurs for the tilted DWs ($\chi(t) \neq 0$) (ref. 32). The velocity remains constant during the current pulse application for the rigid walls ($\chi(t) = 0$): compare the black solid and blue dashed lines in Fig. 2a. Figure 2a,c shows that the velocity decreases while the wall tilting develops. Theoretically, it has been predicted that the time needed to saturate the wall tilting scales with the square of wire width ($w$)[32]. Thus the pulse length required to observe sizable tilting becomes much longer for wider wires. We have studied the wall velocity in wires with $w \sim 5\,\mu$m and $\sim 50\,\mu$m to clarify contribution from the tilting (Supplementary Note 1). For the $\sim 5\,\mu$m wires, we find signatures of wall tilting when longer current pulses are applied (Supplementary Fig. 4). However, for the wider wires, the tilting is not evident (Supplementary Fig. 5). Using typical parameters of the system, we estimate the time it takes to observe the tilting for $w = 50\,\mu$m becomes much longer than the maximum pulse length used here ($\sim 100$ ns). Thus contribution from the wall tilting on $v_{END}$ is negligible when $w \sim 50\,\mu$m.

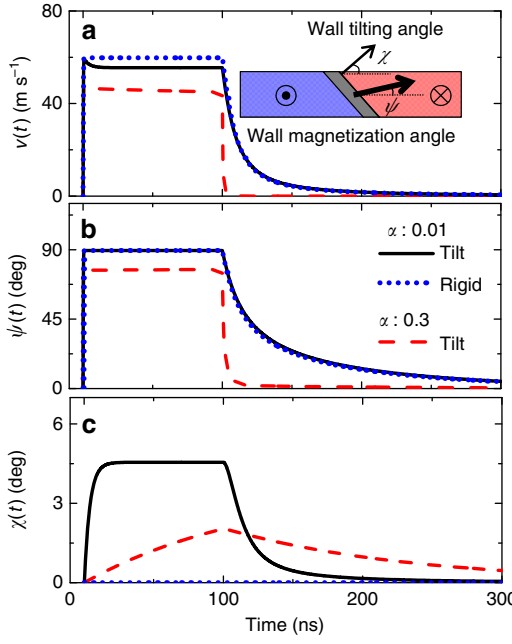

**Figure 2 | One-dimensional model calculations of domain wall (DW) velocity for wires without pinning.** (**a–c**) Instantaneous DW velocity $v(t)$ (**a**), wall magnetization angle $\psi(t)$ (**b**) and wall tilting angle $\chi(t)$ (**c**) for a fixed current density ($J = 0.5 \times 10^8$ A cm$^{-2}$) flowing through the heavy metal layer. The current pulse length is ($t_P$) is 100 ns. Definition of the angles $\psi(t)$ and $\chi(t)$ are illustrated in the inset of **a**. Numerical results for the rigid wall, that is, $\chi(t) = 0$, with low damping ($\alpha = 0.01$) are shown by the blue dotted line whereas results for the tilted walls ($\chi(t) \neq 0$) are shown by the black solid ($\alpha = 0.01$) and red dashed ($\alpha = 0.3$) lines. Parameters used: saturation magnetization $M_S = 1{,}100$ emu cm$^{-3}$, magnetic anisotropy energy $K_{EFF} = 3.2 \times 10^6$ erg cm$^{-3}$, wall width parameter $\Delta = \sqrt{A/K_{EFF}} \sim 6.8$ nm (exchange constant $A = 1.5 \times 10^{-6}$ erg cm$^{-1}$), spin Hall angle $\theta_{SH} = -0.21$, DM exchange constant $D = 0.24$ erg cm$^{-2}$, Gilbert $\alpha = 0.05$ and wire width $w = 5\,\mu$m.

**Determination of the acceleration and deceleration times.**
Thus two different phenomena contribute to the pulse length-dependent wall velocity: the inertia effect that originates from the different acceleration/deceleration times and the wall tilting effect. We first estimate the acceleration and deceleration times using equations (2) and (3) to quantify the inertia effect. The magnetic properties of the films are summarized in Fig. 3. The volume averaged saturation magnetization ($M/V$) and the effective magnetic anisotropy energy ($K_{EFF}$) are plotted against $d$ in Fig. 3a,b. Using these results we calculate the domain wall anisotropy field ($H_K$) and the wall width ($\Delta$). We use $A = 1.5 \times 10^{-6}\, erg\, cm^{-1}$, a typical value for Fe based alloys.

To estimate the acceleration time $\tau_A$ (equation (2)), one needs to know the strength of the spin Hall effective field $H_{SH}$. Here we use the spin Hall magnetoresistance[34–36] to estimate the spin Hall angle, which allows calculation of $H_{SH}$. Interfacial effects, such as the spin memory loss[37,38] or any Rashba–Edelstein related effects[39,40], are neglected for simplicity. First, the resistivity $\rho_N$ of the W layer is obtained by fitting a linear function to the thickness dependence of the resistance inverse $1/R_{XX} \cdot (L/w)$, where $L$ and $w$ are, respectively, the length and width of the wire used to measure the device resistance. The solid line in Fig. 4a shows the fitting result for film set A, which gives $\rho_N \sim 150\, \mu\Omega\, cm$. The resistivity of the W layer for film set A is slightly higher than those reported earlier[17,41,42].

The thickness dependence of the spin Hall magnetoresistance $\Delta R_{XX}/R^Z_{XX}$ is plotted in Fig. 4b. $\Delta R_{XX}$ is the resistance difference of the device when the magnetization of the CoFeB layer points along the film plane perpendicular to the current flow ($R^Y_{XX}$) and along the film normal ($R^Z_{XX}$), that is, $\Delta R_{XX} = R^Y_{XX} - R^Z_{XX}$. The W thickness dependence of spin Hall magnetoresistance can be fitted using the following equation[41–43]: $\frac{\Delta R_{XX}}{R^Z_{XX}} = \theta^2_{SH} \frac{\tanh(d/\lambda_N)}{(d/\lambda_N)(1+\xi)} \left[1 - \frac{1}{\cosh(d/\lambda_N)}\right]$. $\lambda_N$ is the spin diffusion length of the heavy metal (W) layer. $\xi = \rho_N t_{FM}/\rho_{FM}d$ describes the current shunting effect into the magnetic layer ($\rho_{FM}$ is the resistivity of the

magnetic layer: we use $\rho_{FM} \sim 160\, \mu\Omega\, cm$ from our previous study[17]). From the fitting, we obtain $|\theta_{SH}| \sim 0.24$ and $\lambda_N \sim 1.1\, nm$, similar to what has been reported previously[41,42].

The spin Hall effective field ($H_{SH}$) is calculated using the above parameters. If we assume a transparent interface, $H_{SH}$ can be estimated from the following equation[25,44]: $H_{SH} = \theta_{SH} J_N \frac{\hbar}{2eM_s t_F} \left[1 - \frac{1}{\cosh(d/\lambda_N)}\right]$. (If spin memory loss is relevant for the W/CoFeB interface, $H_{SH}$ (and consequently $\tau_A$) will be underestimated.) To calculate the current density $J_N$ that flows into the W layer, we assume two parallel conducting channels (W and CoFeB layers). Calculated $H_{SH}$ is plotted in Fig. 4c for $\sim 5\, \mu m$ and $\sim 50\, \mu m$ wide wires when the pulse amplitude is set to 16 V. The difference in $H_{SH}$ for wires with different widths arises due to the difference in $J_N$. For both cases, however, $H_{SH}$ decreases when $d$ is larger than $\sim 3\, nm$. This is primarily due to the increase in $M_S$ for larger $d$.

To evaluate the deceleration time $\tau_D$ (equation (3)), we must obtain the DM offset field $H_{DM}$. To do so, first the saturation velocity $v_D$ is estimated by the fitting results shown in Fig. 1a-f. Although the velocity is estimated using 10 ns long pulses and equation (1) does not consider any transient effect, we assume that it gives a good estimate of $v_D$ to the first order (see Supplementary Fig. 3 for the justification). $v_D$ is plotted against $d$ in Fig. 4d for both $\sim 5\, \mu m$ and $\sim 50\, \mu m$ wide wires. Next the DM offset field $H_{DM}$ and the DM exchange constant $D$ are calculated using the relations described after equation (1) and plotted against $d$ in Fig. 4e,f, respectively. We find $D$ of $\sim 0.3\, erg\, cm^{-2}$ that is nearly thickness independent and $H_{DM}$ decreasing with increasing $d$ due to the change in $v_D$ and $\Delta$ with $d$ (see refs 17,22 for $D$ of similar heterostructures).

We now have all parameters needed to calculate $\tau_A$ and $\tau_D$. The calculated values are plotted against $d$ in Fig. 4g. In accordance with the results from the 1D model, $\tau_D$ is much larger than $\tau_A$, giving rise to the inertia effect. Note that a significantly large spin memory loss parameter[37] will be required to offset the difference of $\tau_A$ with $\tau_D$. The difference of the two relaxation times, $\tau_D - \tau_A$, provides a good guide for the degree of inertia and is plotted against $d$ in Fig. 4h. $\tau_D - \tau_A$ increases with increasing thickness, reflecting the change in $H_{DM}$ with $d$.

These results can now be compared with the pulse length dependence of the wall velocity shown in Fig. 1g–l. For the thinner W films, we find that $v_{END}$ for shorter pulses do not increase from its long pulse limit, indicating that the inertia is not observable. This is in agreement with the $d$-dependence of $\tau_D - \tau_A$ shown in Fig. 4h except for the device with the thinnest W layer. We note that for even thinner W samples (results not shown in Fig. 4), the domains consist of small grain-like structures and they no longer form a uniform pattern across the device. For such films, domain walls cannot be driven by current.

**Comparison to micromagnetic simulations.** Micromagnetic simulations with realistic pinning are performed to verify the inertia effect and evaluate contribution from the wall tilting (see Supplementary Note 3 for the details). The red squares in Fig. 5a–c show $v_{END}$ versus $t_P$ obtained experimentally for three pulse amplitudes applied to a $\sim 5\, \mu m$ wide wire and $d \sim 3\, nm$. In contrast to $v_{END}$ found in wires with $w \sim 50\, \mu m$ (Fig. 1), $v_{END}$ shows apparent reduction at longer pulses for the narrower wires ($w \sim 5\, \mu m$). The black circles show $v_{END}$ computed using micromagnetic simulations. The simulations are in good agreement with the experimental results. In particular, the simulations can also account for the reduction of $v_{END}$ at longer pulses ($t_P > \sim 20\, ns$): the wall tilting effect becomes evident since the time scale for developing the tilting is close to the pulse length used when $w \sim 5\, \mu m$. Note that the 1D model fails to reproduce

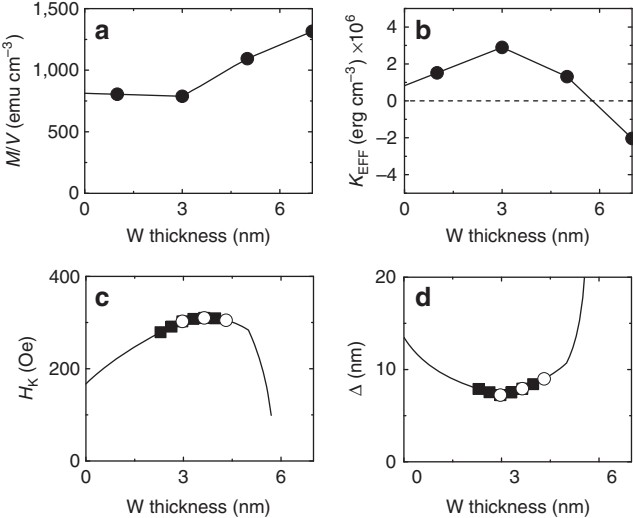

**Figure 3 | Magnetic properties of the films.** (**a,b**) W thickness dependence of the volume averaged saturation magnetization $M/V$ (**a**) and the magnetic anisotropy energy $K_{EFF}$ (**b**). The solid line shows linear interpolation of the data. (**c,d**) DW anisotropy field $H_K$ (**c**) and the wall width parameter $\Delta$ (**d**) calculated from the interpolated data shown in **a** and **b**. The symbols represent values of $H_K$ and $\Delta$ that are used in the calculations presented in Fig. 4 for 5 μm wide wires (open circles) and for 50 μm wide wires (solid square). All results are from film set A.

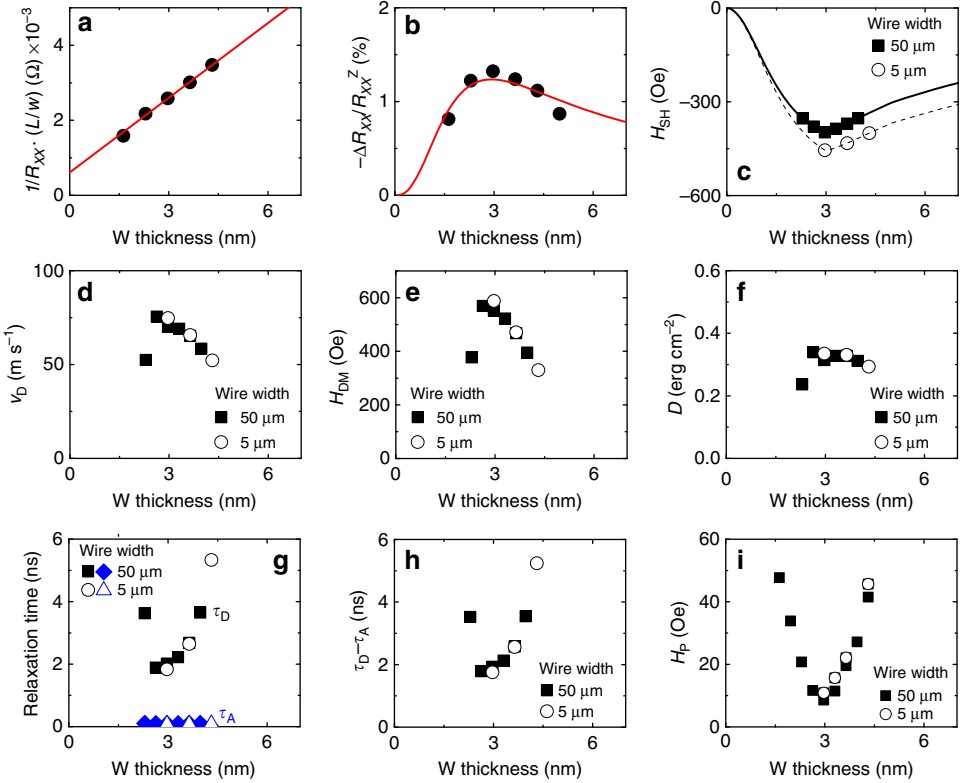

**Figure 4 | Estimated acceleration and deceleration times of domain wall (DW) motion.** (**a**) Normalized inverse resistance ($1/R_{XX}$) as a function of the W layer thickness. $w$ and $L$ corresponds to the width and length of the wire. Data are fitted with a linear function (solid line) to obtain the resistivity of W. (**b**) Spin Hall magnetoresistance $\Delta R_{XX}/R_{XX}^{Z}$ versus W layer thickness. The solid line shows the fitting result. (**c**) Spin Hall effective field $H_{SH}$ calculated using the solid line shown in **b** when a pulse with amplitude of 16 V is applied to the wire. The solid and dashed lines display $H_{SH}$ for $\sim 5\,\mu m$ and $\sim 50\,\mu m$ wide wires, respectively. The symbols represent values of $H_{SH}$ used to calculate the acceleration time shown in **g**. (**d**) The saturation DW velocity ($v_D$) estimated from fitting results of $v_{END}$ vs. pulse amplitude with Equation (1). (**e**,**f**) Calculated DM offset field $H_{DM}$ (**e**) and the DM exchange constant $D$ (**f**). (**g**) W thickness dependence of the acceleration time ($\tau_A$) and the deceleration time ($\tau_D$) estimated using equations (2) and (3), respectively. Black squares (circles): $\tau_D$ for $50\,\mu m$ ($5\,\mu m$) wide wires, blue diamonds (triangles): $\tau_A$ for $50\,\mu m$ ($5\,\mu m$) wide wires. (**h**) Difference of $\tau_D$ and $\tau_A$ plotted against the W layer thickness. (**i**) Average DW propagation field $H_P$ for $\sim 5\,\mu m$ and $\sim 50\,\mu m$ wide wires plotted against the W layer thickness. All results are from film set A.

experimental results at longer pulses as it tends to underestimate the degree of wall tilting. Thus for longer pulses, where the tilting becomes more significant, the velocity reduction is larger for full micromagnetic simulations (Supplementary Figs 8–10).

Figure 5d–f shows the computed average distance ($d_{OFF}$) the DW travels after the current pulse is turned off as a function of $t_P$. $d_{OFF}$ is larger when the pulse length becomes shorter, verifying the inertia effect. Experimentally, we can estimate $d_{OFF}$ using the following relation: $d_{OFF} \sim v(t_P) \cdot \tau_D$. $v(t_P)$ is the instantaneous velocity right before the current pulse is turned off; here we assume it is close to the long pulse limit of $v_{END}$. From the results shown in Fig. 1g–l and Fig. 4g, $d_{OFF}$ is in the range of $\sim 80\,nm$ to $\sim 160\,nm$. This is in good agreement with the results from micromagnetic simulations (Fig. 5d–f).

## Discussion

Although the results presented in Fig. 4g,h indicate that the inertia effect describes the pulse length dependence of $v_{END}$ well, other effects can influence the results. In particular, pinning is not included in deriving the relaxation times $\tau_A$ and $\tau_D$ (equations (2) and (3)) and its influence can be significant in certain occasions. For example, one can imagine that the distance the wall travels after the current pulse is turned off ($d_{OFF}$) will be reduced if the pinning strength becomes significantly larger. Such effect has

been observed in micromagnetic simulations and experiments in certain systems[30].

To study if there is any correlation between the degree of inertia and pinning, the average propagation field $H_P$ versus $d$ is shown in Fig. 4(i) for the $\sim 5\,\mu m$ and $\sim 50\,\mu m$ wide wires. We find that $H_P$ takes a minimum when the domain wall width $\Delta$ is the smallest. Note that it is not always the case that $H_P$ scales with $\Delta$. If pinning plays a dominant role in defining the inertia, we expect to see an inverse relationship between $\tau_D - \tau_A$ and $H_P$. Interestingly, this is not the case here, suggesting that the pinning is not strong enough to influence the inertia significantly.

Finally, equations (2) and (3) and the numerically computed results of the 1D model (Supplementary Fig. 7) indicate that the DW inertia significantly increases when $\frac{\pi}{2}H_{DM}$ approaches $H_K$. This is similar to what was found previously in a different system in which the inertia (that is, the wall mass) increases when $H_K$ approaches zero as the DW makes a transition from a Neel wall to a Bloch wall[1]. Our results demonstrate that one can tune the inertia by material design, wire dimensions and, in some cases, the size of the driving force (for example, current pulses). Large inertia can possibly lead to lower drive current for moving domain walls from pinning sites if one makes use of resonant excitation of domain walls[3]. It is possible to tune the DM interaction in such a way that inertia becomes extremely large or small. These results highlight the unique feature of current driven chiral domain walls.

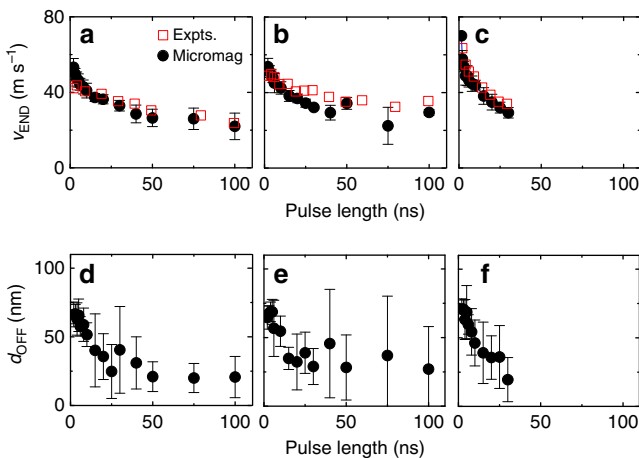

**Figure 5 | Comparison of experiments and micromagnetic simulations**
(**a–c**) Quasi-static domain wall (DW) velocity $v_{END}$ (red squares) measured as a function of pulse length for three different pulse amplitudes:
(**a**) 16 V ($J \sim 0.5 \times 10^8 \, A \, cm^{-2}$) (**b**) 20 V ($J \sim 0.6 \times 10^8 \, A \, cm^{-2}$) and
(**c**) 25 V ($J \sim 0.8 \times 10^8 \, A \, cm^{-2}$). All results are from film set B, wire width is $\sim 5 \, \mu m$ and the W layer thickness $d$ is $\sim 3 \, nm$. The black circles show calculated $v_{END}$ using micromagnetic simulations with two dimensional pinning. The average velocity is obtained from 5 different randomly generated grain patterns. (**d–f**) Average distance DWs travel after the current pulse is turned off ($d_{OFF}$) calculated using micromagnetic simulations. The error bars indicate distribution due to different grain patterns used in the simulations.

## Methods

**Sample preparation.** Films consisting of Sub./W($d$)/Co$_{20}$Fe$_{60}$B$_{20}$(1)/MgO(2)/Ta(1) (units in nanometers) are grown by magnetron sputtering on Si substrates coated with 100 nm thick SiO$_2$. Films are annealed at 300 °C *ex situ* after deposition. Two film sets with nominally the same film structure are made using different sputtering systems. Magnetic and transport properties are slightly different between the two sets. A comparison of the film properties are listed in Supplementary Table 1. Wires, $\sim 5 \, \mu m$ or $\sim 50 \, \mu m$ wide and $\sim 30 \, \mu m$ to $\sim 40 \, \mu m$ long, are patterned using optical lithography and Ar ion etching. A subsequent lift-off process is used to form electrical contacts made of 5 nm Ta|100 nm Au.

**Charcterization of the magnetic properties.** Volume averaged saturation magnetization $M/V$ and magnetic anisotropy energy $K_{EFF}$ of the films are measured using vibrating sample magnetometer (VSM). $M/V$ is obtained by dividing the measured magnetic moment ($M$) by the nominal volume of the magnetic layer ($V$). The nominal volume is equal to the product of the film area (Area) and the thickness ($t_{FM}$) of the magnetic layer, $V = \text{Area} \times t_{FM}$. If a magnetic dead layer exists within the magnetic layer, $M/V$ underestimates the saturation magnetization. For simplicity, here we use $M/V$ for $M_S$ to estimate other quantities. The magnetic easy axis of the films points along the film normal owing to the perpendicular magnetic anisotropy originating from the CoFeB|MgO interface.

**Kerr microscopy imaging.** Motion of domain walls is studied using magneto-optical Kerr microscopy. A voltage controlled pulse generator (Picosecond Pulse Lab, model 10300B) is connected to the device. A pulse or a pulse train consisting of multiple pulses (with fixed pulse length) separated by $\sim 10 \, ms$ is applied to the wire. Before and after the pulse(s) application, Kerr images are captured to determine the distance the domain wall traveled. The bandwidth of the cables and contact probes are DC-40 GHz. Signal transmission is limited by the pulse generator which generates a pulse with $\sim 0.3 \, ns$ rise time and $\sim 0.75 \, ns$ fall time.

**Domain wall velocity in wider wires.** To calculate $v_{END}$ from the Kerr images of the wider ($\sim 50 \, \mu m$ wide) wires, 3–4 rectangular sections, each $\sim 4 \, \mu m$ wide, are defined. The velocity of the wall segment ($\downarrow\uparrow$ walls and $\uparrow\downarrow$ walls) within each section is analysed. The average $v_{END}$ of all sections is shown. Error bars denote standard deviation of $v_{END}$ for all sections (Supplementary Note 1; Supplementary Fig. 5). For the narrower wires ($\sim 5 \, \mu m$ wide) we use one section to calculate $v_{END}$.

**Data availability.** The authors declare that all data supporting the findings of this study are available within the paper and its Supplementary Information files.

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

## Acknowledgements

This work was partly supported by MEXT R & D Next-Generation Information Technology , MEXT Grant-in-Aid for Young Scientists A (23506017) and the Grant-in-Aid for Specially Promoted Research (15H05702). The work by E. M. was supported by project WALL, FP7-PEOPLE-2013-ITN 608031 from European Commission, project MAT2014-52477-C5-4-P from Spanish government, and project SA282U14 from Junta de Castilla y Leon.

## Author contributions

J.T. and M.H. performed the experiments. E.M. performed the micromagnetic simulations. All authors carried out the collective coordinate analysis, contributed to the analyses of the experimental data and writing the manuscript.

## Additional information

**Competing financial interests:** The authors declare no competing financial interests.

