## [Peer Review File · Nature Communications]

Reviewers' comments:

Reviewer #1 (Remarks to the Author):

This manuscript reports a sizable difference between the acceleration time and deceleration time of a magnetic domain wall when it is driven by dampinglike spin-orbit torque. It is found that the dampinglike spin-orbit torque governs the acceleration time whereas the interfacial Dzyaloshinskii-Moriya interaction governs the deceleration time. These results are interpreted as different domain wall inertial mass in the acceleration and deceleration processes, which is the major claim of this paper.

I find that the results are novel and the conclusion is convincing as they are obtained from systematic experimental studies combined with theoretical/numerical studies. As spin-orbit-torque-driven chiral domain wall motion is of considerable interest nowadays, I also find that this paper will be of interest to researchers in the spintronics community.

On the other hand, I'm not pretty sure if the tunable domain wall inertia is an important science. It is now well known that the inertial mass of domain wall is a manifestation of the domain wall tilt angle ψ (as also evidenced by Fig. 2). Effects of dampinglike spin-orbit torque and interfacial Dzyaloshinskii-Moriya interaction on the domain wall tilt angle are also understood fairly well. To my view, thus, the tunable inertia may be useful for applications rather than for an advance in the fundamental science. For instance, the domain wall position after removing the current pulse should be precisely controlled for applications of domain wall devices.

On the overall, this is a fairly nice work. It is impressive that the authors have determined most necessary physical parameters from measurements. I would support publication once the authors have a chance to consider the following minor comments:

1. As mentioned above, it would be nice to include several sentences about the implication of this work for applications.
2. I would suggest to include possible weakness of the model used for the interpretation. For instance, the drift-diffusion model based on the bulk spin Hall effect may be problematic as it does not include spin-orbit coupling effect at the interface (e.g. spin memory loss, Rashba-like band effect, etc). Even though the model is definitely a possible way to choose at this moment, there is no consensus about its validity yet.
3. H_{DM} defined two lines below Eq. (1) may need $\pi/2$.
4. \hbar is the reduced Planck constant.
5. H_K may need to be defined just after Eqs. (2-3).

Reviewer #2 (Remarks to the Author):

This manuscript presents interesting results on the pulse length dependence of current-induced motion of chiral Néel domain walls in W/CoFeB/MgO nanostrips. The authors attribute this dependence to different acceleration and decelerations times at the beginning and the end of a current pulse, associated to a different 'mass' of the domain walls during and after the pulse. These results are in apparent contradiction with previous results on current-induced domain wall motion, some obtained on similar systems. However, the results are rather well reproduced using analytic 1D modeling and 2D micromagnetic simulations. Although the experiments and simulations have apparently been performed carefully, I have several doubts on the physical

interpretation and the implications of the presented results.

1) The authors state that the effective mass of the domain wall is different with the current on and off, given the different acceleration and deceleration times. However, according to Newton's second law of motion, the acceleration/deceleration is proportional to the mass only when the driving force is constant. A change in acceleration time is also obtained for a constant mass and a change in driving force. In the case discussed here, the driving force for the acceleration is the spin Hall torque, while for the deceleration it is the friction against wall motion and the H(DMI) that tends to bring back the wall angle to 0. That the driving forces are different is also clear from equation (S3c) containing terms in H(SH) and u , which are both proportional to the current and thus are very different when the current is on or off. In that case, the mass is not proportional to the acceleration/deceleration times as stated by the authors.

2) The wall angle $\psi(t)$ in Fig. 2(b) is going to 90 degrees as soon as the current pulse is switched on. This should be the case (limited by the precession time) only for current densities for which the domain wall velocity saturates (above about 16V in Fig. 1) and only asymptotically. The acceleration time t_A is calculated using eq. (S3b) with $\psi(\text{eq}) = 0$ or π , but in the case of current densities below the saturation current (which will also increase for increasing D) $\psi(\text{eq})$ and thus t_A will also depend on D . This should be discussed.

3) According to eq.(3) of the main text, t_D and thus the 'mass' as defined by the authors can become infinite, meaning that the domain wall will never stop once the current pulse is switched off. This is partly discussed in the last paragraph before the acknowledgments evoking the possibility of an extremely large mass, but the case of infinite mass, which in principle seems feasible with reasonable material parameters, is not mentioned. The authors should explain why such an unphysical case can not occur.

4) In the introduction, the authors mention several references reporting a DW velocity independent of the pulse length for STT driven domain wall motion. However, in ref. 11 the authors show that also for domain walls driven by Spin Hall torque the velocity is independent of the pulse length, on a system very similar to the one used here. Can the authors explain this discrepancy ?

5) I am puzzled by the results in Fig. 5(d-f) concerning the distance the DW travels after the current is turned off. It is found that $d(\text{OFF})$ is larger when the pulse length becomes shorter, but then it is written that $d(\text{OFF}) \sim v(\text{END}) \times t(D)$. $t(D)$ in eq.(3) does only depend on material parameters and $v(\text{END})$ ($t(p) \rightarrow \infty$) is a constant. So why does $d(\text{OFF})$ depend on pulse length ? It should only depend on the velocity right before the current is switched off and the final wall angle, isn't it ?

Some smaller points :

- The first phrase of the abstract is strange : mass IS a property of an object, so a mass can not be defined by its property...
- On page 8, the authors state that H(DMI) decreases with increasing d due to the change in $v(D)$ and Δ . Do they mean $M(S)$ instead of $v(D)$?
- The authors find that the pinning field $H(p)$ is minimum when the domain wall width has a minimum and the magnetic anisotropy a maximum. Usually the pinning field increases upon increasing the anisotropy and decreasing the wall width. Do the authors know why the contrary is found here ?
- In Appendix B, the authors discuss the effect of Joule heating. They argue that wider wires have a smaller resistance and better cooling. If the cooling takes place through the substrate and the contacts, the width should not influence the heating, for a given current density : the surface is increased but the total current is increased accordingly, the heating/cooling per unit surface should stay identical. On the other hand, when the thickness of W is increased the heating increases for a

given current density, since the total current (and thus the current per unit surface) increases while the cooling surface stays almost the same. The last phrase of the Appendix B is thus not correct, especially considering the word 'unambiguously'.

Reviewer #3 (Remarks to the Author):

The authors studied experimentally the motion of a domain wall in the presence of DM interaction under current pulses and found that the average wall speed depends on the pulse length. They discuss based on a one-dimensional model that the dependence is due to the change of effective mass (inertia) of the wall, and thus the main claim of the manuscript is a tunable inertia of chiral domain wall.

The experimental results would be correct, new and worth publishing. I doubt, however, the theoretical interpretation in term of inertia. As has been well-known, the equation of motion for a domain wall is made up of two equations in the absence of DM interaction, and of three if with DM interaction like in eqs.(S1a-c) in the supplementary material. However, it is also well-known that the wall does not behave like a Newtonian particle with a certain mass unless in very limited cases of small oscillations, as described in detail for instance in Tatara et al, Phys. Rep. 68,213(2008). Here in the present manuscript, I do not see the reason one can linearize the equation of motion as done in eq. (S2). Linearization is not done in Ref. [4] of supplemental material, which the authors' analysis is based on. It is absolutely necessary to provide justification of linearization to discuss or to even define mass. If a justification is explained, the manuscript would be suitable for publication.

In case the linearization was not justified, the manuscript should be written simply to report experimental results, which contain sufficient new results, and not to mention the mass interpretation.

We thank the reviewers for their effort to review the manuscript. The comments from the reviewers were very insightful and have helped us in improving our paper. We have incorporated the reviewers' suggestions in the revised manuscript.

Blue font is the referee's comment and black indicates our reply.

Reviewer #1 (Remarks to the Author):

This manuscript reports a sizable difference between the acceleration time and deceleration time of a magnetic domain wall when it is driven by dampinglike spin-orbit torque. It is found that the dampinglike spin-orbit torque governs the acceleration time whereas the interfacial Dzyaloshinskii-Moriya interaction governs the deceleration time. These results are interpreted as different domain wall inertial mass in the acceleration and deceleration processes, which is the major claim of this paper.

I find that the results are novel and the conclusion is convincing as they are obtained from systematic experimental studies combined with theoretical/numerical studies. As spin-orbit-torque-driven chiral domain wall motion is of considerable interest nowadays, I also find that this paper will be of interest to researchers in the spintronics community.

On the other hand, I'm not pretty sure if the tunable domain wall inertia is an important science. It is now well known that the inertial mass of domain wall is a manifestation of the domain wall tilt angle ψ (as also evidenced by Fig. 2). Effects of dampinglike spin-orbit torque and interfacial Dzyaloshinskii-Moriya interaction on the domain wall tilt angle are also understood fairly well. To my view, thus, the tunable inertia may be useful for applications rather than for an advance in the fundamental science. For instance, the domain wall position after removing the current pulse should be precisely controlled for applications of domain wall devices.

On the overall, this is a fairly nice work. It is impressive that the authors have determined most necessary physical parameters from measurements. I would support publication once the authors have a chance to consider the following minor comments:

1. As mentioned above, it would be nice to include several sentences about the implication of this work for applications.

We thank the referee for the positive and constructive comments. As the referee points out, controlling the inertia associated with domain walls is of particular importance for device applications using the precise positioning of domain walls, for example, in the magnetic Racetrack Memory reported in Parkin *et al*, Science 320, 190 (2008). In general, large inertia is viewed as a disadvantage in terms of device applications since it will hinder fast operation of moving domain walls: one needs to wait for the wall to relax after the current pulse is turned off. However, large inertia can lead to lower drive current if one makes use of resonant excitation of domain walls (see e.g. Thomas *et al.*, Nature 443, 197 (2006)). Our results demonstrate that one can tune the inertia by material design and, in some cases, the size of the driving force (e.g. current pulses). It is possible to tune the DMI in such

a way that inertia becomes extremely large or small. This will enable studies of efficient means to control motion of domain walls by tuning its properties.

As Reviewer #2 noted, it has been reported recently that the wall inertia is almost negligible and the wall looks massless in perpendicularly magnetized thin film heterostructures (e.g. Pt/Co/AlO_x (Ref. 30) and Pt/[Co/Ni] multilayers (Ref. 17)). Our results presented here are in sharp contrast from these reports. We show exactly in what systems the inertia becomes relevant for the current induced motion of domain walls. This will help researchers to design material systems suitable for device applications based on domain walls.

We have added a few comments at the end of the paper to discuss these issues.

2. I would suggest to include possible weakness of the model used for the interpretation. For instance, the drift-diffusion model based on the bulk spin Hall effect may be problematic as it does not include spin-orbit coupling effect at the interface (e.g. spin memory loss, Rashba-like band effect, etc). Even though the model is definitely a possible way to choose at this moment, there is no consensus about its validity yet.

We thank the referee for pointing this out. We agree with the referee that the underlying mechanism of how electrons traverse the W/CoFeB interface is not well characterized, in particular, with regard to the degree of spin memory loss and Rashba-like effect, and requires further investigation. If the drift-diffusion model is not the appropriate model to describe the spin transport across the interface, this will impact the estimation of the damping-like effective field [Fig. 4(c)] and consequently the acceleration time [Fig. 4(g)] of the domain walls. With the parameters used in the paper, the acceleration time is nearly one order of magnitude smaller than the deceleration time. The large difference in the deceleration and acceleration times causes observable inertia effect, as found in the experiments here [Fig. 1(i)-1(l)]. If the damping-like effective field is much smaller, for example, due to spin memory loss, and the acceleration time becomes comparable to the deceleration time, we should find a pulse length independent velocity. However, we estimate that the damping-like effective field needs to be more than ten times smaller than what we have shown in the paper in order to have the acceleration time being comparable to the deceleration time. This will require an extremely large spin memory loss parameter. As micromagnetic simulations with the obtained parameters support our results, we believe that our estimates of the damping-like effective field and the acceleration time are not so off.

With regard to the influence of any Rashba-like effect, it is generally understood that such effect will generate a sizable field-like torque. Unfortunately, we do not have any information on the size of the field-like torque for this system. One can use the harmonic Hall measurements to evaluate the field-like torque; however, due to unidentified reasons, we are not able to extract the current induced effective field in this system (See Ref. 19 for the details). This is why we have relied on the drift-diffusion model to estimate the damping-like effective field. We would like to note that we have performed micromagnetic simulations to study the effect of the field-like torque on the inertia and have found that the effect is almost negligible. Thus we believe that the existence of Rashba-like effect will not alter our conclusion.

We agree with the referee that it is important to include possible weakness(es) of the model used. We have thus included comments in the revised version.

3. H_{DM} defined two lines below Eq. (1) may need $\pi/2$.

We follow the definition of H_{DM} in accordance to the paper from [Ref. 15]. The $\pi/2$ factor pointed out by the referee is included in the LLG equations (Supplementary information, Section II, Eqs. (S1)).

4. \hbar is the reduced Planck constant.

We thank the referee for carefully going through our paper, we have corrected the definition.

5. H_K may need to be defined just after Eqs. (2-3).

Thank you again; we have moved the definition of H_K right after Eqs. (2-3).

Reviewer #2 (Remarks to the Author)

This manuscript presents interesting results on the pulse length dependence of current-induced motion of chiral Neel domain walls in W/CoFeB/MgO nanostrips. The authors attribute this dependence to different acceleration and deceleration times at the beginning and the end of a current pulse, associated to a different 'mass' of the domain walls during and after the pulse. These results are in apparent contradiction with previous results on current-induced domain wall motion, some obtained on similar systems. However, the results are rather well reproduced using analytic 1D modeling and 2D micromagnetic simulations. Although the experiments and simulations have apparently been performed carefully, I have several doubts on the physical interpretation and the implications of the presented results.

1) The authors state that the effective mass of the domain wall is different with the current on and off, given the different acceleration and deceleration times. However, according to Newton's second law of motion, the acceleration/deceleration is proportional to the mass only when the driving force is constant. A change in acceleration time is also obtained for a constant mass and a change in driving force. In the case discussed here, the driving force for the acceleration is the spin Hall torque, while for the deceleration it is the friction against wall motion and the H_{DMI} that tends to bring back the wall angle to 0. That the driving forces are different is also clear from equation (S3c) containing terms in H_{SH} and u , which are both proportional to the current and thus are very different when the current is on or off. In that case, the mass is not proportional to the acceleration/deceleration times as stated by the authors.

We thank the referee for this very important comment. First, we would like to clarify the mathematical analyses described in our paper. If we understand the referee's comment correctly, the main question here is whether the relationship between the relaxation time and the mass holds under a time dependent force. For clarity, Eq. (S2) is rewritten below (the expression is slightly modified)

$$m \frac{\partial^2 q}{\partial t^2} + b \frac{\partial q}{\partial t} = F(t) \quad (\text{R1})$$

This equation can be reduced as the following using the velocity ($v \equiv \frac{\partial q}{\partial t}$):

$$m \frac{\partial v}{\partial t} + bv = F(t) \quad (\text{R2})$$

The force changes with time as the following:

$$F(t) = \begin{cases} F_1 & \text{for } 0 \leq t < t_0 \\ F_2 & \text{for } t \geq t_0 \end{cases} \quad (\text{R3})$$

F_1 and F_2 do not change with time, i.e. they are constant. The pulse length of the driving force (e.g. current) is set to t_0 . The solution to Eq. (R2) is given by

$$v(t) = \begin{cases} A \exp\left(-\frac{t}{\tau}\right) + \frac{F_1}{b} & \text{for } 0 \leq t < t_0 \\ B \exp\left(-\frac{t}{\tau}\right) + \frac{F_2}{b} & \text{for } t \geq t_0 \end{cases} \quad (\text{R4})$$

where $\tau = \frac{m}{b}$, A and B are constants to be determined by the constraints. The following boundary conditions apply to our system:

$$v(t = 0) = 0 \quad (\text{R5a})$$

$$v(t \rightarrow \infty) = 0 \quad (\text{R5b})$$

$$v(t = t_0) = v_0 \quad (\text{R5c})$$

Applying these boundary conditions to Eq. (R4) leads to the following solution:

$$v(t) = \begin{cases} \frac{F_1}{b} \left[1 - \exp\left(-\frac{t}{\tau}\right)\right] & \text{for } 0 \leq t < t_0 \\ v_0 \exp\left(-\frac{t-t_0}{\tau}\right) & \text{for } t \geq t_0 \end{cases} \quad (\text{R6})$$

and once again $\tau = \frac{m}{b}$. As evident, the relationship between the relaxation time and the mass holds even when the force changes at time $t = t_0$.

The referee notes that "the acceleration/deceleration is proportional to the mass only when the driving force is constant". Mathematically speaking, Eq. S2 (or Eq. (R1)) is similar to a damped forced-harmonic oscillator. In such a system, it is not uncommon to have a time dependent force and yet the relation between the mass and the acceleration time remains the same. For example, in a harmonic oscillator, one can apply a force that changes with time in a sinusoidal way and it is common to estimate the mass from the resonance frequency. In a damped system with a time-dependent force, one calculates the mass from the relaxation time, as demonstrated above.

Here, however, we would like to note that the situation is quite different from a typical forced-harmonic oscillator that one encounters in classical mechanics. In solving the equation of motion for a real matter using Newton's second law, one typically assumes a system with *constant mass*. In such system, one can estimate the characteristic time (e.g. resonance frequency for a harmonic oscillator, relaxation time for a damped system) under a time-dependent driving force. For the chiral domain walls discussed here, the mass is not a constant. The *mass changes* when the driving force is turned

on and off. Under such circumstance, it may not be appropriate to use the relationship $\tau = \frac{m}{b}$ to estimate the mass. We have therefore limited the analyses to regimes in which one can regard the mass as a constant (see the discussion in our reply to Reviewer #3 too).

With regard to the analogy of Newton's equation of motion, the domain wall effective mass discussed here should not be interpreted as the mass in real world. As stated in the seminal book from Malozemoff and Slonczweski (Ref. 1), "... this mass is not a real mass but only an effective mass whose properties are merely analogous to real matter because of the analogy between the wall-dynamics equations of motion and Hamilton's equations of motion". The concept of effective mass is useful to describe transient effects like the domain wall inertia discussed here. At the moment, we cannot think of real world analogue that responds to a force in the way that the chiral wall does here.

Section IIC in the revised supplementary is added to discuss this issue.

2) The wall angle $\psi(t)$ in Fig. 2(b) is going to 90 degrees as soon as the current pulse is switched on. This should be the case (limited by the precession time) only for current densities for which the domain wall velocity saturates (above about 16V in Fig. 1) and only asymptotically. The acceleration time t_A is calculated using eq. (S3b) with $\psi_{eq} = 0$ or π , but in the case of current densities below the saturation current (which will also increase for increasing D) ψ_{eq} and thus t_A will also depend on D. This should be discussed.

We thank the referee for pointing this out. First, the acceleration time is calculated assuming an equilibrium domain wall magnetization angle $\Psi_{eq} \sim \pi/2$ or $-\pi/2$, not 0 or π as the referee stated (perhaps this is a mistake in the referee's comment). We assume the main question here is whether the acceleration time derived in Eq. (2) holds when the current density becomes smaller than the "saturation current density", i.e. when Ψ_{eq} does not reach $\pi/2$ or $-\pi/2$.

In order to clarify this issue, we have numerically calculated the relaxation times (τ_A and τ_D) by solving Eqs. (S1a)-(S1c) under different current densities and compared the results with those estimated using Eqs. (2) and (3). The results are presented below in Fig. R1, which is the same as that newly added in the supplementary material (as Fig. S6). The parameters used are similar to those described in Fig. 2 (with Gilbert damping $\alpha=0.05$). For simplicity, we assume the tilt angle χ to be zero here.

Figures R1(a,b) show the instantaneous velocity as a function of time when a 100 ns long current pulse is applied. The acceleration and deceleration times are obtained by fitting the velocity vs. time using Eq. (R6). As the time variation of the wall angle is the main source of the relaxation effects, we have calculated and fitted Ψ vs. time using similar exponential functions: the calculated and fitted curves are shown in Figs. R1(c,d). The left and right panels show calculation results using different current densities. Note that the equation of motion (Eq. (S2) or Eq. (R1)) and its solution (Eq. (R6)), i.e. the exponential function, are valid only when the wall angle Ψ is close to $\pm\pi/2$ when the current is on and 0 (or π) when it is off. We therefore limit the fitting range to which the exponential function can be applied: deviation of Ψ from its equilibrium value is set to be less than ~ 20 deg.

First, from the fitting, we obtain the saturation velocity (v_D) and the corresponding equilibrium wall angle (i.e. Ψ_{eq}) when current is applied. These quantities are plotted in Fig. R1(e) and R1(f) using the solid symbols. (As a guide to the eye, the solid line in Fig. R1(e) shows the numerically calculated saturation velocity at the end of the current pulse ($t=100$ ns).) Figures R1(e) and R1(f) show that Ψ_{eq}

decreases with decreasing current density, resulting in a smaller velocity at lower current. In the parameter set used here, a considerably decrease in Ψ_{eq} and v_D occurs when the current density is smaller than $\sim 0.2 \times 10^8 \text{ A/cm}^2$.

Fig. R1. (a-d) Instantaneous DW velocity $v(t)$ (a,b) and the wall magnetization angle $\psi(t)$ (c,d) for a fixed current density of $J = 0.02 \times 10^8 \text{ A/cm}^2$ (a,c) and $J = 0.6 \times 10^8 \text{ A/cm}^2$ (b,d) flowing through the heavy metal layer. The current pulse length is (t_p) is 100 ns. Fit to data in appropriate ranges using Eq. (S6) are shown by the red and blue solid lines. (e-h) Current density J dependence of saturation velocity (e), the equilibrium wall angle (f), the acceleration time (g) and the deceleration time (h) when the current is turned on. Results are obtained by the fitting process described in (a-d). The blue solid line in (g) and (h) are the analytical solutions provided in Eqs. (2) and (3), respectively. Parameters used: $M_S = 1100 \text{ emu/cm}^3$, $K_{EFF} = 3.0 \times 10^6 \text{ erg/cm}^3$, $\Delta = \sqrt{A/K_{EFF}} \sim 7.0 \text{ nm}$ ($A = 1.5 \times 10^{-6} \text{ erg/cm}$), $\theta_{SH} = -0.21$, $D = 0.24 \text{ erg/cm}^2$, $\alpha = 0.05$ and $w=5 \text{ }\mu\text{m}$.

The corresponding acceleration (τ_A) and deceleration (τ_D) times are shown in Figs. R1(g) and R1(h). We show τ_A and τ_D obtained by fitting the velocity vs. time (black squares) and Ψ vs. time (red circles) and compare those to the values calculated using Eqs. (2) and (3) (blue solid line). We find that the numerical calculations and the analytical solutions of the acceleration time τ_A are in good

agreement even for small current densities at which Ψ_{eq} is much smaller than $\pi/2$. These calculations show that the estimation of τ_A using Eq. (2) is valid at smaller current although its derivation assumes $\Psi_{eq} \approx \pm\pi/2$.

The numerical calculations of the deceleration time (Fig. R1(h), solid symbols) show that τ_D varies little with the current density. This is in good agreement with Eq. (3) which dictates that τ_D is constant against the current density.

We have added Section IIC and Fig. S6 in the revised supplementary to address this issue.

3) According to eq.(3) of the main text, t_D and thus the 'mass' as defined by the authors can become infinite, meaning that the domain wall will never stop once the current pulse is switched off. This is partly discussed in the last paragraph before the acknowledgments evoking the possibility of an extremely large mass, but the case of infinite mass, which in principle seems feasible with reasonable material parameters, is not mentioned. The authors should explain why such an unphysical case cannot occur.

Again, we thank the referee for this important question. As the referee points out, Eqs. (2) and (3) indicate that there will be a condition at which the mass becomes infinite. An infinite mass occurs when the net domain wall anisotropy field becomes zero. Mathematically, we consider such condition is allowed (this is also discussed by Malozemoff and Slonczewski).

Experimentally, however, it will be nearly impossible to probe the infinite mass as thermal activation and non-zero pinning will obscure the measurements to probe the large mass. For example, the "infinite deceleration time" is not possible due to pinning, i.e. the wall will be stopped by pinning even though the LLG equation indicates that the wall will continue to decelerate for an infinite time. Similarly, an immobile domain wall due to the infinite mass can still move due to thermal activation, again prohibiting its detection.

4) In the introduction, the authors mention several references reporting a DW velocity independent of the pulse length for STT driven domain wall motion. However, in ref. 11 the authors show that also for domain walls driven by Spin Hall torque the velocity is independent of the pulse length, on a system very similar to the one used here. Can the authors explain this discrepancy ?

We believe the main differences between our results and those of Ref. 11 are the damping and the DMI. The damping value estimated for Pt/CoNi is ~ 0.15 (Ref. 12), which is at least 3 times larger than the system studied here (W/CoFeB/MgO). For Pt/Co, even higher values of damping, 0.3~0.5, (see e.g. Schellekens *et al*, APL 102, 082405 (2013), Miron *et al.*, Nat. Mater. 10, 419-423 (2011)) have been reported previously. As the deceleration time is inversely proportional to the damping, it becomes difficult to probe the inertial effect in such large damping systems. (The role of the damping is clearly shown in Fig. 2 for two extreme damping values of 0.01 and 0.3.)

In addition, the DM exchange field (H_{DM} in the paper) is likely much larger than the magnetostatic anisotropy field (H_K) in the Pt/CoNi samples studied in Ref. 11. In order to observe a large inertia effect, τ_D has to be much larger than τ_A , and thus H_{DM} has to be close to H_K in size as indicated by Eq. (3). Recent studies show that the DM exchange field is of the order of ~ 1000 Oe at the Pt/Co interface (Refs. 14, 21), which is likely much larger than H_K given that the saturation magnetization M_S is smaller than the samples studied here (H_K is proportional to M_S). Our paper clearly shows in what systems one would expect a large inertial effect in the motion of domain walls.

5) I am puzzled by the results in Fig. 5(d-f) concerning the distance the DW travels after the current is turned off. It is found that $d(\text{OFF})$ is larger when the pulse length becomes shorter, but then it is written that $d(\text{OFF}) \sim v(\text{END}) \times t(\text{D})$. $t(\text{D})$ in eq.(3) does only depend on material parameters and $v(\text{END})$ is a constant. So why does $d(\text{OFF})$ depend on pulse length ? It should only depend on the velocity right before the current is switched off and the final wall angle, isn't it ?

This issue is partly discussed in the supplementary material. As the referee notes, τ_D is defined by material parameters, and the terminal velocity (the velocity when the current pulse is cut off) is constant if the domain wall tilting is neglected. However, in the micron-sized wires studied here, the tilting effect is not negligible and it can influence the terminal velocity, see the black solid line in Fig. 2(a). For the wires studied in Fig. 5 ($\sim 5 \mu\text{m}$ wide), the time needed to develop the tilting (τ_χ , Eq. (S6)) is similar in magnitude with the pulse length used, i.e. $\tau_\chi \sim 20\text{-}100$ ns. Thus the terminal velocity drops as the pulse length is increased, and consequently d_{OFF} decreases with increasing pulse length. Moreover, we find that the degree of tilting is larger than the simple 1D case (Fig. 2) once we take into account the two dimensional profile of the domain wall using micromagnetic simulations. The two dimensional pinning landscape promotes larger tilting, and thus we find an even larger reduction in d_{OFF} at longer pulses.

In contrast, the tilting effect is in general negligible for the wider wires ($\sim 50 \mu\text{m}$ wide, Figs. 1,3,4) since the time to develop tilting is much longer than the pulse length: τ_χ scales with the square of the wire width (see Eq. (S6)). We therefore believe the results from the wider wires presented in Figs. 1,3,4 are more straightforward to understand. However, since the tilting effect can be important in some systems, we have included the results in Fig. 5.

Some smaller points:

- The first phrase of the abstract is strange : mass IS a property of an object, so a mass can not be defined by its property...

Thank you for pointing this out, we have corrected the sentence.

- On page 8, the authors state that $H(\text{DMI})$ decreases with increasing d due to the change in $v(\text{D})$ and Δ . Do they mean $M(\text{S})$ instead of $v(\text{D})$?

We thank the referee for carefully reading our paper. Indeed, H_{DM} is a function of M_{S} , as noted in the text. However, one can rewrite the expression for H_{DM} to show that it is a function of v_{D} and Δ , which is described in the line above. Here we have employed the latter.

- The authors find that the pinning field $H(\text{p})$ is minimum when the domain wall width has a minimum and the magnetic anisotropy a maximum. Usually the pinning field increases upon

increasing the anisotropy and decreasing the wall width. Do the authors know why the contrary is found here ?

We agree with the referee that the dependence of H_p on the anisotropy energy is in contrast to what is expected empirically. In many systems, we also find the pinning field being proportional to the perpendicular magnetic anisotropy energy.

Although this may be a coincidence, the W thickness dependence of H_p coincides with that of the difference in the deceleration and acceleration times $\tau_D - \tau_A$, a quantity proportional to the effective wall mass. These results indicate that, the heavier the wall is, the larger the propagation field. It remains to be confirmed whether the effective wall mass plays any role in defining the propagation field.

- In Appendix B, the authors discuss the effect of Joule heating. They argue that wider wires have a smaller resistance and better cooling. If the cooling takes place through the substrate and the contacts, the width should not influence the heating, for a given current density: the surface is increased but the total current is increased accordingly, the heating/cooling per unit surface should stay identical. On the other hand, when the thickness of W is increased the heating increases for a given current density, since the total current (and thus the current per unit surface) increases while the cooling surface stays almost the same. The last phrase of the Appendix B is thus not correct, especially considering the word 'unambiguously'.

We thank the referee for pointing this out. We agree with the referee that the heating/cooling per unit surface is not going to improve with increasing the wire width. The description is rephrased.

Reviewer #3 (Remarks to the Author):

The authors studied experimentally the motion of a domain wall in the presence of DM interaction under current pulses and found that the average wall speed depends on the pulse length. They discuss based on a one-dimensional model that the dependence is due to the change of effective mass (inertia) of the wall, and thus the main claim of the manuscript is a tunable inertia of chiral domain wall.

The experimental results would be correct, new and worth publishing. I doubt, however, the theoretical interpretation in term of inertia. As has been well-known, the equation of motion for a domain wall is made up of two equations in the absence of DM interaction, and of three if with DM interaction like in eqs.(S1a-c) in the supplementary material. However, it is also well-known that the wall does not behave like a Newtonian particle with a certain mass unless in very limited cases of small oscillations, as described in detail for instance in Tataru et al, Phys. Rep. 68,213(2008). Here in the present manuscript, I do not see the reason one can linearize the equation of motion as done in eq. (S2). Linearization is not done in Ref. [4] of supplemental material, which the authors' analysis is based on. It is absolutely necessary to provide justification of linearization to discuss or to even define mass. If a justification is explained, the manuscript would be suitable for publication.

In case the linearization was not justified, the manuscript should be written simply to report experimental results, which contain sufficient new results, and not to mention the mass interpretation.

We thank the referee for pointing out this critically important issue, it has helped us to review the analyses we have used.

First, in Ref. [4] of the supplementary material (Martinez *et al.*, J. Appl. Phys. 115, 213909 (2014)), the focus was at quantities in equilibrium (e.g. the saturation velocity v_D given after Eq. (1)). At equilibrium, one can assume that the wall magnetization angle and the velocity stay constant, i.e. $d\psi/dt=0$ and $dq/dt=const$, and thus without introducing the linearization one can analytically calculate certain quantities. In contrast, here we need to calculate quantities that vary in time to study the transient effect. When the wall angle and the velocity are varying with time, we cannot use the constraints above and thus the linearization is needed to simplify the equations in order to obtain the analytical solutions.

In our previous report (Ref. 19), we have used the linearization process to describe our experimental results. The analyses were based on the models described in Refs. 14 and 15, one of the first reports on spin Hall effect driven chiral domain walls. (Many of the important concepts on the motion of chiral domain walls are discussed in Ref. 23).

If we understand the referee's point correctly, our main task here is to clarify the validity of the linearization process used to derive Eqs. (2) and (3). First, we would like to note that Ref. 6, one of the first papers to show that there is a significant inertia in STT driven domain walls, uses the linearization process to derive the relaxation time. Our model follows this approach: we attempt to find approximate expressions for the relaxation (acceleration and deceleration) times by linearizing the LLG equation of chiral domain walls driven by the spin Hall effect.

The difference between the system used here and that of Ref. 6 is the degree of changes in the wall angle when current is turned on. In Ref. 6, the wall angle ψ stays close to 0 (or π) at low current. Thus the linearization of the LLG equation can be justified at any instance. However, here the wall angle ψ changes between 0 (or π) to $\pm\pi/2$ when the current is tuned on and off. As it is not appropriate to linearize the equation under such change in the wall angle, we have limited our solutions to two extreme cases: i.e. when the wall magnetization angle approaches ~ 0 (or π) and $\pm\pi/2$.

To justify the approximation used in the paper, we have numerically calculated the relaxation times and compared them to the analytical solutions (Eqs. (2) and (3)) provided in the paper. (The calculations are carried out also in response to Reviewer #2's comment.) The comparison is shown in Figs. R1(g) and R1(h). The solid symbols show the acceleration and deceleration times obtained by fitting the velocity or the wall magnetization angle vs. time using an exponential function. Note that the equation of motion (Eq. (S2) or Eq. (R1)) and its solution, i.e. the exponential function (Eq. (R6)), are valid only when the wall angle ψ is close to $\pm\pi/2$ when the current is on and 0 (or π) when it is off. We therefore limit the fitting range to which the exponential function (Eq. (R6)) can be applied; i.e. to a time range in which deviation of ψ from its equilibrium value is less than ~ 20 deg.

Within such fitting range, we find that the numerical calculations of the acceleration time τ_A are in good agreement with the analytical solution. The numerical calculations of the deceleration time (Fig. R1(h), solid symbols) are ~ 10 - 20% smaller than the analytical estimate. The little current density dependence of the deceleration time is evident both for the numerical and analytical calculations. From these results, we consider the expressions given in Eqs. (2) and (3) provide good estimates of

the relaxation times, justifying the linearization of the LLG equation. In response to Reviewer #2's comments, the relationship $\tau = \frac{m}{b}$ should hold when the change in the wall angle is small (i.e. for the two extreme cases in which the wall magnetization angle is close to ~ 0 (or π) or $\pm\pi/2$) since the effective wall mass can be considered a constant under such constraint. Thus one may use Eq. (4) to estimate the effective wall mass from the relaxation time as a first guess.

We would like to comment on the difficulty in fitting the relaxation process with the exponential function. The difficulty arises since the wall mass, or the relaxation time, continues to evolve during the transient processes (this is also briefly discussed in our reply to Reviewer #2' comments). Under such condition, it is not appropriate to use a function with a single relaxation time. Ideally, to describe the relaxation process, one would need to use a relaxation time that is a weighted average of the processes involved. We consider finding a proper function to describe the relaxation process is beyond the scope of this paper.

We may ignore the physics behind the relaxation processes, as discussed above, and fit the entire curve without limiting the fitting range to obtain an *effective relaxation time* of the numerical results. However, in many cases, the exponential function does not fit well the numerical results (see Fig. R2(a-d) below), which is consistent with the discussion above, i.e. a single relaxation time does not capture the transient processes.

In the following, we compare the *effective relaxation time* (τ_A^{EFF} and τ_D^{EFF}) with that obtained using the fitting with the limited Ψ range (Fig. R1(g) and 1(h)). Here we use the *effective relaxation time* obtained from the best fitting: the fitting of the wall angle Ψ vs. time at higher current (Fig. R2(d)). From the fitting, we obtain $\tau_A^{EFF} \sim 0.11$ ns and $\tau_D^{EFF} \sim 10.8$ ns. τ_A^{EFF} is in good agreement with that of τ_A obtained using fitting with the limited Ψ range ($\tau_A \sim 0.10$ ns in Fig. R1(d)). In contrary, the τ_D^{EFF} is a factor of ~ 2 smaller than that of τ_D in Fig. R1(d), $\tau_D \sim 18.3$ ns. We consider the fitting with the limited Ψ range provides a first order estimate of the relaxation time.

Fig. R2. (a-d) Instantaneous DW velocity $v(t)$ (a,b) and the wall magnetization angle $\psi(t)$ (c,d) for a fixed current density of $J = 0.02 \times 10^8$ A/cm² (a,c) and $J = 0.6 \times 10^8$ A/cm² (b,d) flowing through the heavy metal layer. The current pulse length is (t_p) is 100 ns. Fit to data using Eq. (S6) are shown by the red and blue solid lines. The fitting range is 0-100 ns and 100-200 ns for the red and blue lines, respectively. Parameter used for the numerical calculations are the same as those in Fig. R1.

As the referee questioned the validity of the linearization process, we have also tried a 2nd order expansion of Ψ in the LLG equation (Eqs. (S1)). The solution of such 2nd order Ψ expanded-LLG equation is not a simple exponential function: the form becomes more complicated. Although the solution fits the numerical results better than those of Eq. (R6), they do not provide intuitive and analytic picture of the physics involved. We thus consider the analytical expressions obtained by linearizing the LLG equations (Eqs. (2) and (3)) give the best interpretation of the results within a reasonable accuracy.

We have added Section IIC and Fig. S6 in the revised supplementary to address this issue.

Changes made to the revised manuscript (highlighted by red font in the main text)

- The first sentence of the abstract is changed as the following: "The mass of any object is defined by its *material parameter* its property or by the environment" (in response to Reviewer #2's comment)
- "reduced" is inserted before the "Planck constant" in page 3 (in response to Reviewer #1's comment)
- " M_S is the saturation magnetization" is included in page 3
- Definition of H_K is inserted after Eq. (3) in page 5 (in response to Reviewer #1's comment)
- A comment on the evolving relaxation time and the effective wall mass is included in page 6 (in response to Reviewer #2 and #3's comments)
- Assumption made for the interfacial spin orbit effects are described in page 7 (in response to Reviewer #1's comment)
- The influence of the interfacial effects on the estimation of the relaxation times is discussed in page 8 (in response to Reviewer #1's comment)
- A comment on the possible relation between the propagation and the effective wall mass is included in page 10 (in response to Reviewer #2's comment)
- The impact of our findings is briefly summarized at the end of the paper in page 11 (in response to Reviewer #1's comment)
- References 31, 37, 38 and 39 are added (in response to Reviewer #1's comment)
- Minor changes are made throughout the text.

Changes made to the supplementary material

- Section IIC and Fig. S6 are added to describe the validity of the linearized model (in response to Reviewer #2 and #3's comments)
- Description related to the Joule heating is rephrased in section IB (in response to Reviewer #2's comment)

Reviewers' comments:

Reviewer #4 (Remarks to the Author):

Tunable inertia of chiral magnetic domain walls

Jacob Torrejon, Eduardo Martinez, Masamitsu Hayashi

The authors studied current induced domain wall motion in W/CoFeB/MgO/Ta large wires. They report a pulse length dependent velocity and more precisely a velocity increase at short current pulses. They explain this result by the contribution of two phenomena: the wall tilting effect (for sufficient long current pulses vs the width of the wire) and the different acceleration/deceleration times. They focus their message on this fast acceleration and slow deceleration that they studied using both a 1D collective coordinate model as well as micromagnetic simulation. They found that the acceleration is governed by the spin Hall effect while the deceleration is governed by the DMI. Using the analogy of the equation of the DW motion with the Newton's equation of motion they claim that "the effective mass of a chiral Néel DW is much lighter when the wall is driven by a current via the SHE compared to that at rest" and that the inertia of chiral DWs can be tuned based on their results.

I find that the experimental results are interesting and novel. Compared to previous results, the low damping of their material as well as the relatively small DMI (HDM not so far from HK) allow them to observe a pulse length dependent velocity.

Their analysis in terms of different acceleration/deceleration time is convincing. However as, pointed by referee 1 their analysis does not involved any new physics: it is now well known that the inertia of a domain wall is a manifestation of its deformation [ref 6, 7], and in the particular case of perpendicular magnetized material, of the rotation of its core magnetization, measured by the ψ angle. This is actually shown by the authors who reproduce their experimental results using standard 1D analytical model and micromagnetic simulations including DMI and SHE.

I am less convinced by the use of the concept of mass (as pointed by both referee 2 and referee 3). As the authors write in their article as well as in their answer to the referees: "the domain wall effective mass discussed here should not be interpreted as the mass in real world". Moreover, they analysis in terms of one relaxation time (exponential decay) linked to the mass of the DW is only correct for a certain range of ψ angle values. Since the inertia is related to the time variation of the ψ angle, the analogy they use to introduce the mass results in a mass that is not a constant. This point was raised by referee 2 and the answer is not completely satisfactory: the authors show the calculation of a very simple and particular case to conclude that, as far as one relaxation time τ can be defined, the linear relationship between τ and the mass holds. However the problem here is that the relaxation cannot be entirely described by an exponential with one τ and that in consequence the "mass" cannot be a constant but is time dependent. This conclusion is underlined by the authors: "the mass is not a constant for the chiral domain walls discussed here" (answer to referee 2).

Despite this, the authors insist on the concept of mass to present their results, writing for example in their abstract that: "[...] A magnetic domain wall (DW) is a topological object which can be treated as a classical particle with a well-defined mass and momentum". I do not understand this, as it seems to be contradictory with both their results and their answer to referee 2.

To conclude, the experimental results presented here are interesting and novel as they show a pulse dependent velocity due to the different acceleration and deceleration time, observable in materials with low damping and relatively small DMI (HDM not so far from HK) and they would deserve publication. However, I find that the manuscript tries to oversell the results using the

concept of mass while the physics is not novel and entirely described by standard equations. I would like the authors to clarify this point before supporting publication.

We thank the reviewer for his/her effort to review the manuscript. We appreciate the reviewer's comments which have helped us to improve our paper. We have incorporated the reviewer's suggestions in the revised manuscript.

Blue font is the referee's comment and black indicates our reply.

Reviewer #4 (Remarks to the Author):

The authors studied current induced domain wall motion in W/CoFeB/MgO/Ta large wires. They report a pulse length dependent velocity and more precisely a velocity increase at short current pulses. They explain this result by the contribution of two phenomena: the wall tilting effect (for sufficient long current pulses vs the width of the wire) and the different acceleration/deceleration times. They focus their message on this fast acceleration and slow deceleration that they studied using both a 1D collective coordinate model as well as micromagnetic simulation. They found that the acceleration is governed by the spin Hall effect while the deceleration is governed by the DMI. Using the analogy of the equation of the DW motion with the Newton's equation of motion they claim that "the effective mass of a chiral Néel DW is much lighter when the wall is driven by a current via the SHE compared to that at rest" and that the inertia of chiral DWs can be tuned based on their results.

I find that the experimental results are interesting and novel. Compared to previous results, the low damping of their material as well as the relatively small DMI (HDM not so far from HK) allow them to observe a pulse length dependent velocity.

First, we are pleased to hear that the reviewer considers our experimental results "interesting and novel". Indeed, the pulse length dependent velocity found in the system here is due to the small damping and the DM exchange field (H_{DM}) compensating the wall anisotropy field (H_K).

Their analysis in terms of different acceleration/deceleration time is convincing. However as, pointed by referee 1 their analysis does not involved any new physics: it is now well known that the inertia of a domain wall is a manifestation of its deformation [ref 6, 7], and in the particular case of perpendicular magnetized material, of the rotation of its core magnetization, measured by the ψ angle. This is actually shown by the authors who reproduce their experimental results using standard 1D analytical model and micromagnetic simulations including DMI and SHE.

We are happy to know that the referee agrees with our analyses of the experimental results using the acceleration/deceleration times. Although many of our results can be described using the 1D analytical model and micromagnetic simulations, we consider there are many new physics revealed by the experimental results and the modelling that were not known before. Experimentally, this is the first paper to report pulse length dependent current driven velocity of chiral domain walls. We demonstrate that this is due to the difference in the acceleration and deceleration times, a characteristic that is not expected for a typical classical object. Using analytical and numerical models, we show the acceleration and deceleration times are associated with different material parameters, the former being determined by the spin Hall angle of the heavy metal layer whereas the latter being dominated by the DM exchange field. Such difference in the relaxation times allows one to tune the inertia of chiral domain walls, either by

material design or in some cases, by the current itself. We believe that these observations provide new insight to spin Hall torque driven chiral domain walls.

I am less convinced by the use of the concept of mass (as pointed by both referee 2 and referee 3). As the authors write in their article as well as in their answer to the referees: “the domain wall effective mass discussed here should not be interpreted as the mass in real world”. Moreover, they analysis in terms of one relaxation time (exponential decay) linked to the mass of the DW is only correct for a certain range of ψ angle values. Since the inertia is related to the time variation of the ψ angle, the analogy they use to introduce the mass results in a mass that is not a constant. This point was raised by referee 2 and the answer is not completely satisfactory: the authors show the calculation of a very simple and particular case to conclude that, as far as one relaxation time τ can be defined, the linear relationship between τ and the mass holds. However the problem here is that the relaxation cannot be entirely described by an exponential with one τ and that in consequence the “mass” cannot be a constant but is time dependent. This conclusion is underlined by the authors: “the mass is not a constant for the chiral domain walls discussed here” (answer to referee 2).

Despite this, the authors insist on the concept of mass to present their results, writing for example in their abstract that: “[...] A magnetic domain wall (DW) is a topological object which can be treated as a classical particle with a well-defined mass and momentum”. I do not understand this, as it seems to be contradictory with both their results and their answer to referee 2.

We thank the referee for pointing out this issue on the wall mass. We agree with the referee that the sentence in the abstract "A magnetic domain wall (DW) is a topological object which can be treated as a classical particle with a well-defined mass and momentum" can cause confusion. In the system studied here the wall mass changes when current is turned on, thus the term "well-defined" may not be appropriate. We have changed the abstract to address the referee's concern.

We would like to note that wall mass defined in Eq. (4) of the previous version is valid within the framework of linearized equation of motion of the domain walls. Thus if one accepts that the acceleration and deceleration times are valid, the wall mass defined in Eq. (4) is equally valid. To justify our conclusion, we have provided in the supplementary material (Fig. S7) additional numerical calculations of the 1D model. These calculations confirm that significant inertia results when the conditions correctly specified by the referee are met (i.e. low damping, H_{DM} close to H_K). Thus even without the solution of the linearized 1D model, our conclusion holds. On the other hand, we consider the analytical solutions (Eqs. (2) and (3)) of the linearized equations are useful in understanding the underlying physics, and thus we prefer to keep the current figures.

To conclude, the experimental results presented here are interesting and novel as they show a pulse dependent velocity due to the different acceleration and deceleration time, observable in materials with low damping and relatively small DMI (HDM not so far from H_K) and they would deserve publication. However, I find that the manuscript tries to oversell the results using the concept of mass while the physics is not novel and entirely described by standard equations. I would like the authors to clarify this point before supporting publication.

We thank the referee for acknowledging the importance of our paper. Our intent to use the effective wall mass is not to oversell the results but rather to articulate the discussion that is acceptable to a broader audience. We considered it may be more intuitive to describe the results in terms of effective wall mass since "mass" is a quantity that is easy to picture.

However, as the referees expressed concern of using the effective wall mass to interpret the experimental results, we have decided to rephrase the abstract/intro and move all discussions related to the wall mass, including Eq. (4) in the previous version, into the supplementary information. The abstract and introduction are changed to avoid using terms related to wall mass but we attempt to keep the content clear to a broad readership. We hope that these changes address the concerns raised by the referees.

Changes made to the revised manuscript

- As per referee's comment, all discussions related to the wall mass, including Eq. (4) in the previous version, is moved to the supplementary material. The abstract and introduction are also rephrased to accommodate these changes.
- Supplementary information: Fig. S7 and section S2D are added to support the conclusion drawn by the analytical solutions.
- Minor changes are made throughout the text to improve the paper.